# Conservation Significance of the Rare and Endangered Tree Species, *Trigonobalanus doichangensis* (Fagaceae)

Ling Hu [1,2], Xin-Gui Le [3], Shi-Shun Zhou [4], Can-Yu Zhang [5], Yun-Hong Tan [4], Qiang Ren [1], Hong-Hu Meng [1,4,*], Yupeng Cun [6,7,*] and Jie Li [1,8,*]

1 Plant Phylogenetics and Conservation Group, Center for Integrative Conservation, Xishuangbanna Tropical Botanical Garden, Chinese Academy of Sciences, Kunming 650223, China
2 University of Chinese Academy of Sciences, Beijing 100049, China
3 Yangjifeng National Nature Reserve of Jiangxi Province, Guixi 335400, China
4 Southeast Asia Biodiversity Research Institute, Chinese Academy of Sciences, Naypyidaw 05282, Myanmar
5 Yunnan Normal University, Kunming 650500, China
6 Pediatric Research Institute, Ministry of Education Key Laboratory of Child Development and Disorders, National Clinical Research Center for Child Health and Disorders, China International Science and Technology Cooperation Base of Child Development and Critical Disorders, Chongqing Key Laboratory of Translational Medical Research in Cognitive Development and Learning and Memory Disorders, Children's Hospital of Chongqing Medical University, Chongqing 400014, China
7 Germplasm Bank of Wild Species, Kunming Institute of Botany, Chinese Academy of Sciences, Kunming 650201, China
8 Center of Conservation Biology, Core Botanical Gardens, Chinese Academy of Sciences, Mengla 666303, China
* Correspondence: menghonghu@xtbg.ac.cn (H.-H.M.); cunyp@cqmu.edu.cn (Y.C.); jieli@xtbg.ac.cn (J.L.)

**Abstract:** *Trigonobalanus doichangensis* is a rare and endangered species with important evolutionary value and extremely small populations. We investigated the genetic diversity of *T. doichangensis* to provide information on its effective preservation. We used genotyping-by-sequencing (GBS) technology to assess the genetic diversity, genetic structure and gene flow of the six populations of *T. doichangensis*. Analysis of SNPs indicated that there was high genetic diversity in the ML and XSBN populations of *T. doichangensis*. $F_{ST}$ values showed moderate genetic differentiation among the populations of *T. doichangensis*. Meanwhile, admixture, principal components and gene flow analyses indicated that the populations of *T. doichangensis* are not genetically separated in accordance with their geographical distributions. Habitat destruction and excessive exploitation may have led to a low gene flow, which has in turn resulted in the differences in seed and seedling morphological traits among populations. Based on these findings, we recommend that *T. doichangensis* be conserved through in situ approaches and artificial seedlings, including preservation of each extant population. Particularly, the ML and XSBN populations have high diversity and more ancestral information, so these two populations should be considered as conservation priorities, and seeds should be collected to obtain germplasm and increase the genetic diversity.

**Keywords:** biodiversity; conservation; *T. doichangensis*; genetic structure; gene flow; conservation priority

## 1. Introduction

The arrival of the Anthropocene is bringing an unprecedented challenge to the biodiversity that is essential to humans and enhancing the value of many of the benefits of nature to human beings [1]. Biodiversity is vital to human well-being and an important pillar of ecosystem balance [2]. Since the adoption of the Convention on Biological Diversity (CBD), various countries have made several achievements and contributions in different areas with the aim of maintaining biodiversity and safeguarding the Earth's resources, but biodiversity loss and ecosystem degradation remain important problems [3]. China is rich in biological resources and contains several global biodiversity hotspots. Its

unique geographical distribution and climatic zones have shaped a wide range of diverse biotypes and it is considered a major center of origins and diversity [4]. However, with rapid economic growth and dramatic population increases, conflicts between economic development and biological conservation have come to the fore, and China is becoming one of the countries with the most serious threats to biodiversity [5]. An assessment of over 35,000 species of wild higher plants in China found that approximately 10.84% of species are in a threatened state [6]. Continued loss of biodiversity and chronic environmental conflicts suggest that conservation communities, including authorities, institutes and scholars, need to re-examine the assumptions and practices upon which the conservation endeavor has been founded, particularly in tropical China, which harbors hyper-diverse plant species [7,8]. In recent years, the National Implementation Plan for Rescuing and Conserving China's Plant Species with Extremely Small Populations (PSESP) was formulated following the presentation of the concept of PSESP [9]. In China, it is these species that are the focus of the PSESP program, species with extremely small populations are precisely the ones most in need of urgent conservation action [10]. The goal of the PSESP program, developed and implemented in 2005 in Yunnan Province, is to try to secure a long-term future for these plants in peril [11]. From then on, some taxa of PSESP have been brought into focus; e.g., *Aristolochia delavayi* [12], *Camellia huana* [13], *Cycas panzhihuaensis* [14], *Horsfieldia tetratepala* [15], *Rhododendron meddianum* [16], *Rhododendron pubicostatum* [17], etc. The introduction of the PSESP program provides a target for the conservation of endangered plants in China; however, the endangerment mechanism and threat status of species are important indicators for species conservation [18]. Only by clarifying the actual threat status, endangerment mechanism and genetic diversity of species can the direction of conservation be targeted and conservation measures clarified [18].

Genetic diversity is one of the key pillars of biodiversity, and high genetic diversity increases wild plants ability to survive and reduces the risk of extinction for species, thus allowing the prediction of the fitness of species through the study of genetic diversity [19,20]. In addition, genetic differentiation and gene flow are important elements in understanding the evolutionary and adaptive potential of populations. Previous studies have shown that many endangered plants tend to have low genetic diversity and high genetic differentiation due to overexploitation and habitat destruction [15,21,22]. Inbreeding can occur between populations as a result of fewer individuals existing, leading to the gradual accumulation of deleterious mutations between populations, producing severe geographical isolation and placing the survival of populations at risk [23]. The loss of endangered species habitats also disrupts gene flow between populations and leads to population isolation, a negative effect that seriously threatens the development and diversity structure of species, while high gene flow reduces the incidence of inbreeding and population differentiation by increasing the exchange of genetic material between populations [22]. However, there are some endangered species that exhibit a genetic structure with high genetic diversity and low genetic differentiation [24,25]. In general, plant genetic diversity is influenced by seed dispersal, reproductive systems, life history, geographic range and evolutionary history. Thus, species with long life histories, extensive seed dispersal routes and outcrossing will likely retain more genetic variation, allowing them to resist environmental change, but over-exploitation and habitat destruction, as well as limitations within the species themselves, will lead to population contraction and a dramatic reduction in the number of individuals. Then, over time, inbreeding and genetic drift will increase population differentiation and reduce the gene flow between populations, leading again to a high degree of differentiation between populations [15,20,26]. Therefore, the combination of various aspects in the exploration of species diversity levels can help us to propose more accurate conservation strategies [22].

The fields of ecological and conservation genetics have developed significantly in recent decades thanks to the use of molecular markers to investigate organisms in their natural habitat and to evaluate the effect of anthropogenic disturbances [27]. However, previous studies mainly focused on genetic diversity, population structure, genetic characterization

based on chloroplast DNA sequence, single-copy nuclear genes and microsatellite markers. Few studies have used next-generation sequencing (NGS) to investigate single nucleotide polymorphisms (SNPs), which have become the marker of choice for determining population structure as they are abundant, stable in the genome and can be accurately scored [28]. Among the NGS methods used in population genetics, the genotyping-by-sequencing (GBS) approach is suitable for population studies, germplasm characterization, breeding and trait mapping in diverse organisms [29]. GBS methods offer major advantages for population genomics thanks to their capacity to screen thousands of polymorphisms throughout the genome and highlight the full range of evolutionary histories (variation in drift, selection, recombination, mutation) and consequences for genetic variation [27]. This method is based on genome-wide sequencing of loci adjacent to conservative restriction sites, which makes it possible to obtain thousands of homologous loci for a set of species with no prior genome data [30].

*Trigonobalanus* shows many ancestral features typical of the Fagus-Quercus taxa [31,32]. *Trigonobalanus doichangensis* belongs to *Trigonobalanus*, which includes *T. doichangensis*, *T. verticillata* and *T. excelsa*. *T. doichangensis* is restricted to south Yunnan, China—i.e., Ximeng, Menglian, Puer, Lancang and Cangyuan—and Chiang Mai in northern Thailand (Figure 1) [32,33]. *T. doichangensis* and *T. verticillata* are distributed across tropical Asia, while *T. excelsa* is found in Columbia, Central America, which is important to understand the phylogeny and biogeography of Fagaceae in relation to continental drifts, climatic shifts and the past global environment [8].

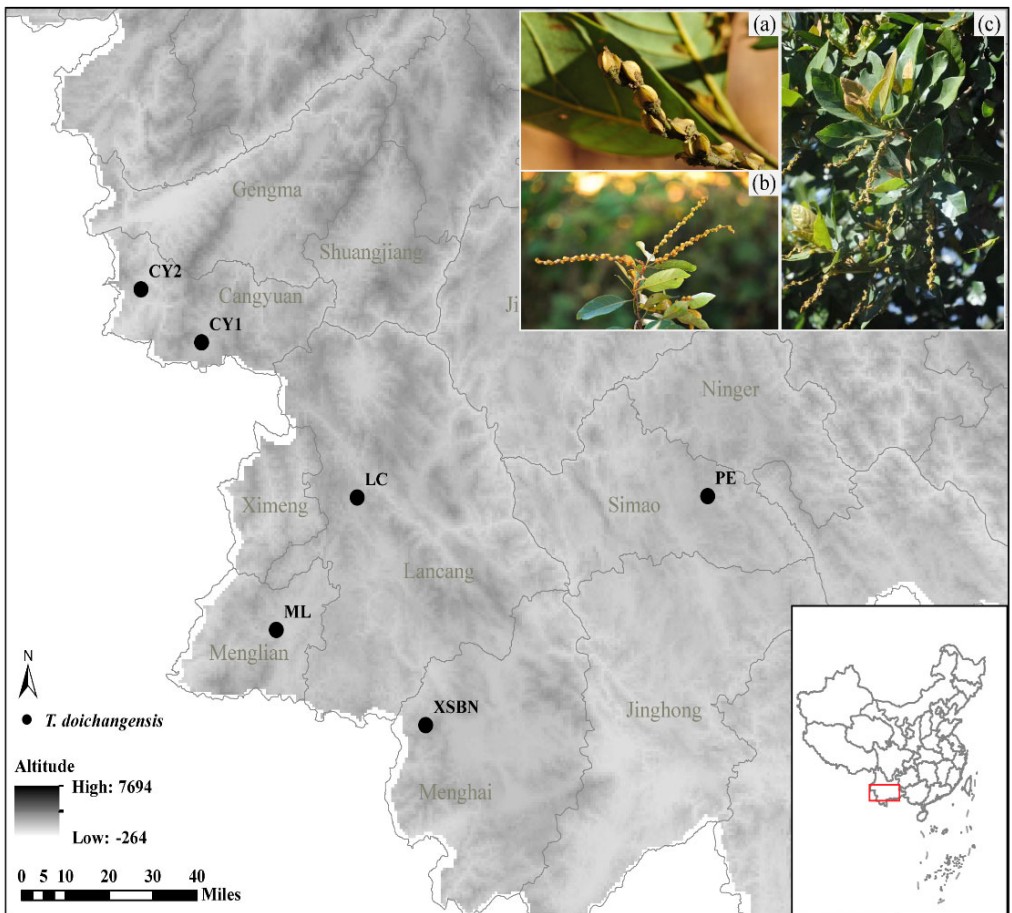

**Figure 1.** Sampling locations of *T. doichangensis*. (**a**,**b**) pistillate inflorescences of *T. doichangensis*; (**c**) leaf of *T. doichangensis*.

*T. doichangensis* has been felled for firewood, house building and agricultural tools [34]. Due to this heavy exploitation, along with habitat degradation caused by clearing areas

for agriculture, *T. doichangensis* has been pushed to the verge of extinction [35]. Despite being a scientifically important tree with endangered status, the species is rare in China and, thus, poorly represented in herbaria; the species has been proposed as the second ranked endangered plant for national protection in the China Species Red List [36].

For the conservation of the rare and endangered *T. doichangensis* tree in southwest China, the extent of the genetic variation in seed and seedling traits suggested seed collection of this species for ex situ conservation and species restoration [34]. Historically, most studies on ecological and conservation genetics have relied on a small number of putatively neutral molecular markers [27]. Similarly, the population genetics of *T. doichangensis* have long been assessed using random amplified polymorphic DNA (RAPD), which suggested that habitat degradation, overexploitation and reproductive barriers are most likely to be the factors threatening the species [35]. To date, there has been little research on the population structure and genetic diversity of *T. doichangensis* based on genome-wide SNP information, although this information is important for the development of meaningful conservation management strategies for this species.

In this study, we used GBS to investigate relationships between populations of *T. doichangensis* based on 39 samples and determine the genetic diversity, population structure and genetic distance and gene flow of *T. doichangensis* in Yunnan, China. Our aim in this study was to develop meaningful conservation strategies and suggestions for *T. doichangensis*.

## 2. Materials and Methods

### 2.1. Sample Collection and Genotyping-by-Sequencing

We collected 39 individuals from six populations of *T. doichangensis* from its distribution areas in China (Figure 1; Table 1). The total genomic DNA of *T. doichangensis* was extracted using a Plant Genomic DNA Kit (Tiangen Biotech, Beijing, China). A GBS library was prepared by Shanghai Majorbio Bio-pharm Technology Co., Ltd. Double digestion was performed using *Mse*I and *Taqα*I then sequenced on an Illumina Hi-seq sequence platform with paired-end (PE) 150 sequencing.

**Table 1.** Sample information for *T. doichangensis*.

| No. | Populations | No. | Locations | Longitude | Latitude | Elevation (m) |
|-----|-------------|-----|-----------|-----------|----------|---------------|
| 1 | XSBN | 9 | Padianliangzi, Xishuangbanna, Yunnan | 100°04′58″ | 22°02′32″ | 2038 |
| 2 | LC | 6 | Donghuixiang, Lancang, Yunnan | 99°48′50″ | 22°43′20″ | 1400 |
| 3 | CY1 | 5 | Chengbian, Cangyuan, Yunnan | 99°14′28″ | 23°10′18″ | 1482 |
| 4 | CY2 | 8 | Nanbancun, Cangyuan, Yunnan | 99°01′04″ | 23°19′27″ | 1482 |
| 5 | PE | 5 | Dazhaishuiku, Puer, Yunnan | 101°03′43″ | 22°45′44″ | 1564 |
| 6 | ML | 6 | Dengzhanzhai, Menglian, Yunnan | 99°32′30″ | 22°18′47″ | 1153 |

### 2.2. SNP Calling and Quality Filter

We used FastQC (https://www.bioinformatics.babraham.ac.uk/projects/fastqc, accessed on 14 October 2021) and Fastp (http://github.com/OpenGene/fastp, accessed on 14 October 2021) for quality control of the raw data, which was undertaken by Shanghai Majorbio Bio-pharm Technology Co., Ltd. skrTools (https://github.com/sharkLoc/skrTools, accessed on 7 November 2021) was used to calculate the number of reads, the GC content and the average quality of the data (Q30) for the GBS of *T. doichangensis*.

We employed the *process_radtags* module in STACKS v 2.55 [37] to filter low-quality reads and ambiguous bases. Due to STACKS need for consistent sequence lengths, the sequence lengths were trimmed to 135 bp to ensure subsequent analysis: -r -c -q -t 135. Clean reads were aligned to the reference genome of *Castanea mollissima* [38] with default parameters in BWA-MEM v 0.7.17 [39] and sorted via SAMtools v 1.8 [40], which provided the final BAM file for each sample. We built loci from the aligned paired-end data with the *gstacks* module in STACKS. Finally, we used the *populations* module in STACKS for SNP

calling, specifying a genotyping rate of at least 75% of individuals within populations and a minimum number of three populations per locus, outputting the first SNP per locus.

To ensure downstream analysis, we conducted sample quality control on the SNPs, using VCFtools v 0.1.16 [41] to view the missing rates of samples and loci, randomly identifying samples with a missing rate of more than 70% according to the population and merging the BAM file to increase the signal of variation sites using SAMtools-merge. We reran the *gstacks* module and *populations* module in STACKS for the SNP calling; the parameters were the same as before, and randomly sampling was conducted in R v 4.1 [42].

The vcf file was further filtered in VCFtools using the following parameters: SNPs with missing rates greater than 20% were removed (–max-missing 0.8); SNPs with minor allele counts lower than 3 (–mac 3) were retained; SNPs with average depths greater than or equal to 3 were reserved (–minDP 3); minor allele frequencies lower than 0.05 were also removed (–maf 0.05).

### 2.3. Calculation of Genetic Diversity Parameter and AMOVA

The parameters of population genetic diversity included private alleles (PAs), expected heterozygosity (*He*), observed heterozygosity (*Ho*), pairwise genetic distance ($F_{ST}$), inbreeding coefficient ($F_{IS}$) and nucleotide polymorphism (*Pi*), which were calculated using the *populations* module in STACKS; the $F_{ST}$ value was calibrated using the *p*-value.

In order to determine whether genetic variation was present within the populations, we used Arlequin v3.5.2 [43] for analysis of molecular variance (AMOVA), where groups were defined based on the results of the structural clustering. The corresponding file formats were converted using PGDspider v 2.1.1.5 [44].

### 2.4. Isolation by Distance

$F_{ST}$ was used as the genetic distance between populations. Geographic distances between populations were calculated according to their geographic coordinate information using the *geosphere* package [45]. The relationship and significant differences between genetic distance and geographic distance were assessed with the Mantel test using the *ade4* package [46] in R, and 9999 permutations were performed to determine significance.

### 2.5. Population Structure and Gene Flow

Population structure was analyzed in terms of the different numbers of ancestors (*K*) in Admixture v 1.3 [47], where *K* was chosen between 1 and 7 (assumed populations + 1) [48]. The most likely number of genetic clusters was computed with a fivefold cross-validation (CV) error. Finally, population structure was visualized via the *pophelper* package [49] in R. Principal components analysis (PCA) clearly showed hierarchical clustering for each population. Based on the previous results, PLINK v 1.9 [50] was used to generate files for PCA analysis, and then the *ggplot2* package [51] in R software were used for the PCA display. Discriminant analysis of principal components (DAPC) was used to infer the population structure by determining the number of clusters observed without prior knowledge, and the principal components from the PCA obtained from the data dimensionality reduction were used to perform linear discriminant analysis (LDA) [52]. Since DAPC is more robust than PCA [53,54], we added DAPC to verify the results from Admixture and the PCA. We used the *adegenet* package [55] in R to plot DAPC results. TreeMix v 1.12 [56] was used to infer splitting and mixture patterns among populations of *T. doichangensis*. Migration events (m) from 0 to 6 were specified between populations, and 10 iterations per event were tested. For this analysis, "-global" and "-se" options were used to calculated the standard errors, the "-noss" parameter prevented overcorrection of the sample size and the "-bootstrap -k 1000" parameter was used to build a maximum likelihood tree (ML) by resampling blocks of 1000 SNPs [57]. The *OptM* package [58] in R was used to determine the optimal number of migration edges using the output file from the TreeMix, and the R script *plotting_funcs.R* (https://github.com/joepickrell/pophistory-tutorial/blob/master/example2/plotting_funcs.R, accessed on 6 March 2022) was used to visualize the migration

results. Then, we ran the *threepop* module and *fourpop* module in TreeMix to analyze the f3-statistics and f4-statistics with -k 1000. In the f3-statistics (C; A, B), a significantly negative statistic for the f3 score (Z-score < −3) indicated that C was a mixture of A and B. In the f4-statistics (A, B; C, D), a significantly negative statistic for the f4 score (Z-score < −3) indicated gene flow between populations related to A and D or B and C, and a significantly positive statistic for the f4 score (Z-score > 3) indicated gene flow between populations related to A and C or B and D [59].

## 3. Results

### 3.1. Genotyping-by-Sequencing and Quality Control

All 39 samples of *T. doichangensis* were sequenced using GBS, which produced 412,410,372 clean reads and the average value for the quality scores of the reads (Q30) was 88.52%, while the GC content was 41.74%. After running the *process_radtags* module, a total 377,488,916 clean reads were obtained, in which the GC content was 41.66% and the Q30 was 89.06%.

SNP calling was conducted according to the method described in Section 2.2. Since the missing rates in the sampling locations XSBN and ML were relatively high (Figure 2), the XSBN and ML populations were randomly sampled and merged as new BAM files for SNP calling. A total of 38,182 variant sites were obtained, and 7591 filtered SNPs were used for downstream analysis.

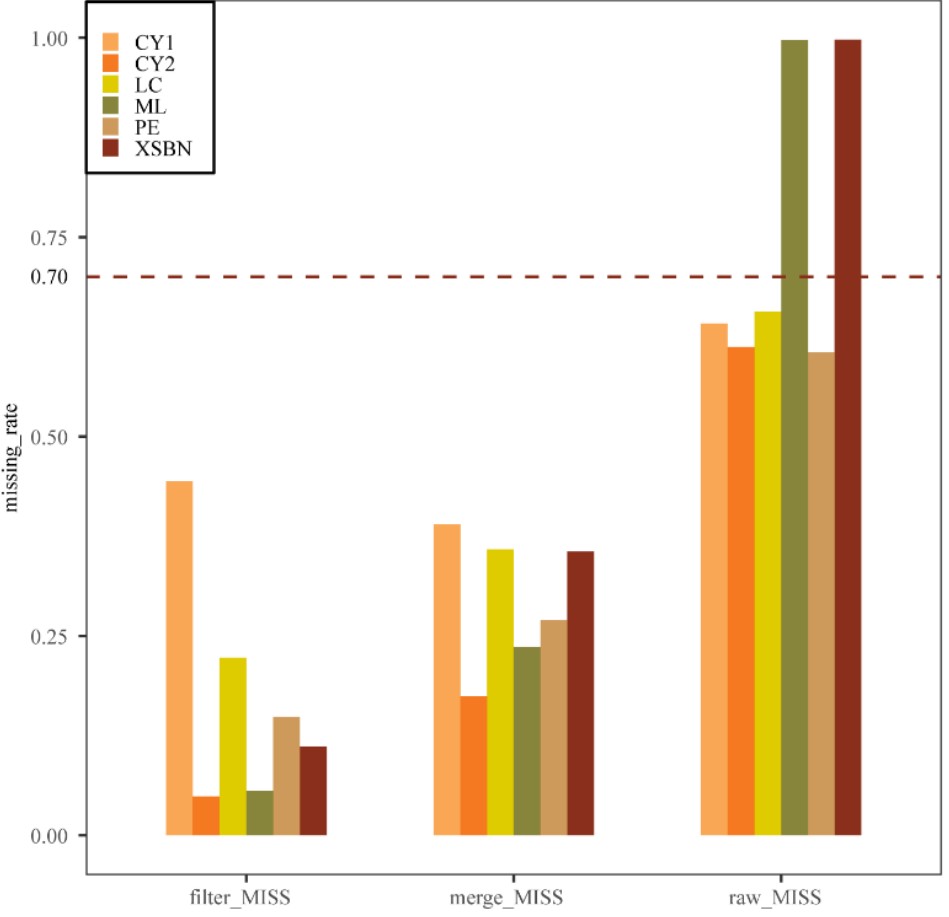

**Figure 2.** The missing SNP rate for each population without merging (raw_MISS); The missing SNP rate for each population after merging (merge_MISS); and the missing SNP rate for each population after filtering (filter_MISS).

### 3.2. Genetic Diversity and Genetic Differentiation

The results for the genetic diversity parameters of *T. doichangensis* are summarized in Table 2 and Figure 3. The observed heterozygosity (*Ho*) ranged from 0.3129 to 0.5571; the

expected heterozygosity (*He*) ranged from 0.2232 to 0.3059; nucleotide diversity (*Pi*) ranged from 0.2504 to 0.3343; the inbreeding coefficient (*F_IS*) ranged from −0.4213 to −0.0879 (Table 2). The ML population showed the highest genetic diversity (*Ho* = 0.5571; *He* = 0.3059; *Pi* = 0.3343; Table 2) followed by the XSBN population, and the PE population showed the lowest genetic diversity (*Ho* = 0.3129; *He* = 0.2232; *Pi* = 0.2504; Table 2). Furthermore, the XSBN population had the most private alleles, and the CY1 population had the fewest private alleles, while the *F_IS* values of the six populations were negative, indicating that there was no inbreeding within the six populations.

**Table 2.** Genetic diversity parameters for *T. doichangensis* populations based on GBS.

| Populations | PA | *Ho* | *He* | *Pi* | *F_IS* |
|---|---|---|---|---|---|
| XSBN | 642 | 0.5410 | 0.3044 | 0.3235 | −0.4213 |
| LC | 10 | 0.3169 | 0.2383 | 0.2621 | −0.1022 |
| CY1 | 3 | 0.3129 | 0.2335 | 0.2637 | −0.0879 |
| CY2 | 48 | 0.3150 | 0.2430 | 0.2598 | −0.1033 |
| PE | 32 | 0.3129 | 0.2232 | 0.2504 | −0.1151 |
| ML | 284 | 0.5571 | 0.3059 | 0.3343 | −0.4166 |

Notes: PA, private allele; *Ho*, observed heterozygosity; *He*, expected heterozygosity; *Pi*, nucleotide diversity; *F_IS*, inbreeding coefficient.

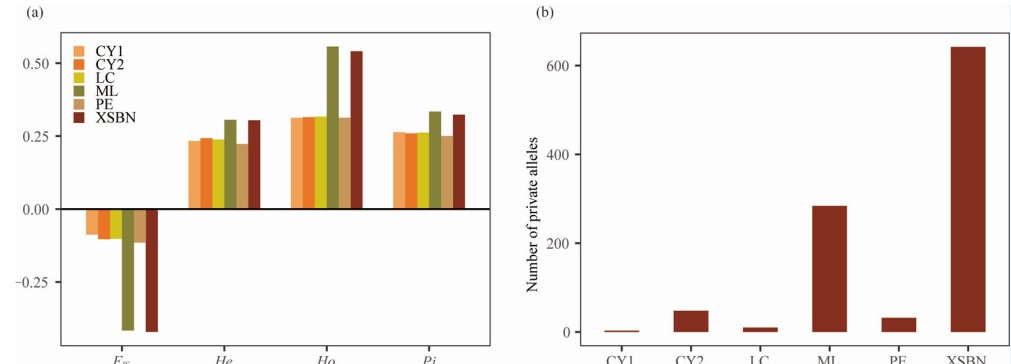

**Figure 3.** The results for the diversity parameters of *T. doichangensis* populations. (**a**) Expected heterozygosity (*He*), observed heterozygosity (*Ho*), inbreeding coefficient (*F_IS*) and nucleotide polymorphism (*Pi*); (**b**) number of private alleles (PAs).

The values of $F_{ST}$ ranged from 0.0462 to 0.1402 (Table 3, Figure 4a) with an average of 0.0921, and the average $Nm = (1 − F_{ST})/4 * F_{ST} = 2.7194$. The results showed that there was genetic differentiation and gene flow between populations. In addition, the CY1 and CY2 populations showed the lowest $F_{ST}$ ($F_{ST}$ = 0.0462) and the ML and XSBN populations showed the largest $F_{ST}$ ($F_{ST}$ = 0.1402), indicating that only 4.62% of the variation occurred between the CY1 and CY2 populations and most (95.38%) occurred within populations. CY1 and CY2 were geographically closest and in the same region (Figure 1; Table 3), which may also have been the reason for the low genetic differentiation between CY1 and CY2. However, after we analyzed the correlation between genetic distance and geographic distance, we found that there was no significant correlation between genetic distance and geographic distance (Figure 4b). The AMOVA results also indicated that variation occurred within populations rather than among populations (Table 4).

**Table 3.** Genetic distance and geographic distance between populations of *T. doichangensis*. Below diagonal: genetic distance among populations ($F_{ST}$). Above diagonal: geographic distance between populations (km).

| Populations | XSBN | LC | CY1 | CY2 | PE | ML |
|---|---|---|---|---|---|---|
| XSBN | - | 80.5984 | 152.6067 | 179.8030 | 128.7604 | 63.4062 |
| LC | 0.0980 | - | 77.1407 | 105.5585 | 128.2083 | 53.4627 |
| CY1 | 0.1014 | 0.0573 | - | 28.4602 | 192.1101 | 100.4373 |
| CY2 | 0.1121 | 0.0561 | 0.0462 | - | 218.5402 | 124.7333 |
| PE | 0.1210 | 0.0779 | 0.0829 | 0.0779 | - | 164.1128 |
| ML | 0.1402 | 0.0957 | 0.1004 | 0.1013 | 0.1133 | - |

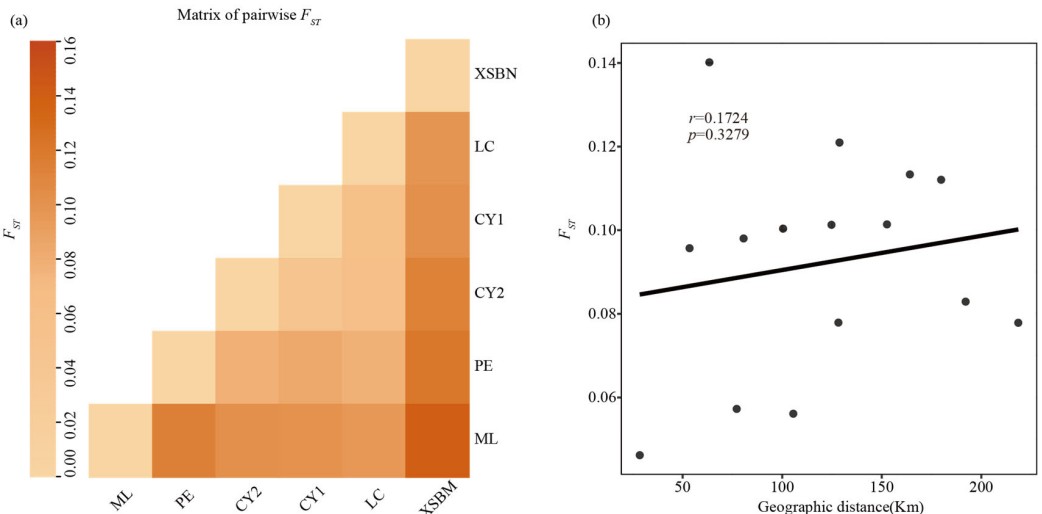

**Figure 4.** Pairwise $F_{ST}$ for *T. doichangensis* populations (**a**) and the correlation between genetic distance and geographic distance (**b**).

**Table 4.** Analysis of molecular variance (AMOVA) of the genetic variation in *T. doichangensis*.

| Source of Variation | d.f. | Sum of Squares | Variance Components | Percentage of Variation |
|---|---|---|---|---|
| Among groups | 2 | 3210.467 | 49.1535 | 9.97 |
| Among populations within groups | 3 | 1617.129 | 8.8010 | 1.79 |
| Within populations | 72 | 31312.635 | 434.8977 | 88.24 |
| Total | 77 | 36140.231 | 492.8522 | - |

Notes: Groups are optimal results of genetic structure. d.f., degree of freedom.

### 3.3. Genetic Structure of T. doichangensis

We performed PCA, DAPC and genetic structure analysis to understand the genetic structure of each population. Fivefold cross-validation (CV) was chosen for the genetic structure analysis, and the results showed that the optimal number of ancestral components for the *T. doichangensis* populations was 3 (Figure S1). When K = 3, XSBN and ML populations formed cluster 1 (red) and cluster 3 (purple), respectively, except for a few admixed individuals. The LC, CY1, CY2 and PE populations formed a mixed-component cluster (cluster 2, green) (Figure 5c). Moreover, the results of the PCA analysis also confirmed the results from Admixture (Figure 5a). The PCA results based on 7591 SNPs showed that principal components (PC1, PC2) represented 45.3% and 29.4% of all variation, respectively (Figure 5a). The XSBN and ML populations from Xishuangbanna and Menglian each formed a single cluster, while the LC, CY1, CY2 and PE populations formed close clusters (Figure 5a). In addition, we increased the number of principal components and further analyzed the first three principal components. We found that, with three principal compo-

nents, the PE population was further separated (Figure S2). The PE population is located in Puer city, Yunnan, China, and is separated from the XSBN, LC, CY1 and CY2 populations on both sides of the Lancang River. Therefore, we assumed that the geographical barrier was the reason for the separation of the PE population. We used the first three PC axes and two discriminant axes to perform DAPC, using the Bayesian information criterion (BIC) to define the best cluster (Figure S3). The first eigenvalue was 369.2 and the second eigenvalue was 151.3. The results are the same as in the genetic structure analysis and the first two principal components of the PCA (Figure 5b).

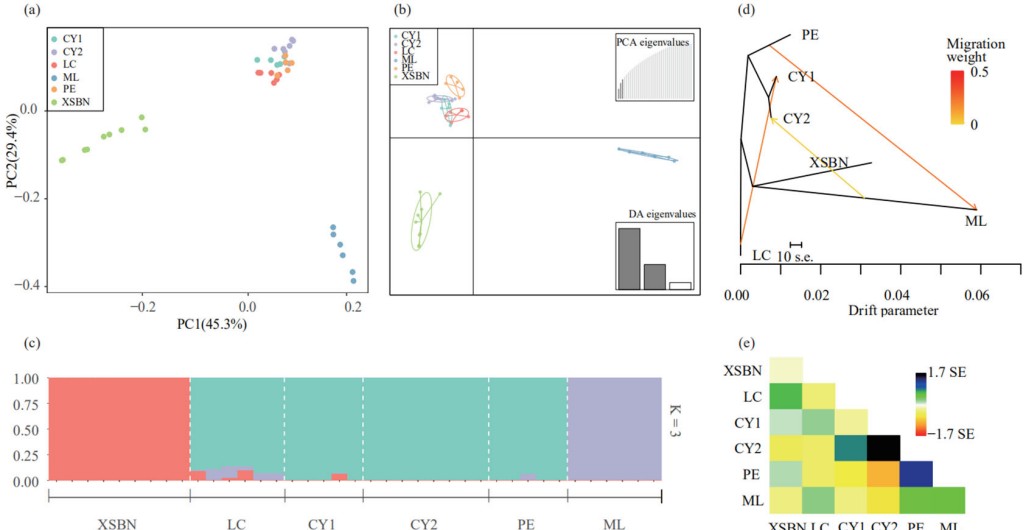

**Figure 5.** (**a**) Principal components analysis (PCA) plot generated for *T. doichangensis*. (**b**) Discriminant analysis of principal components (DAPC) plot generated for *T. doichangensis*. (**c**) Genetic structure bar plots at K = 3 from clustering analysis among six populations using Admixture. (**d**) Unrooted maximum likelihood phylogenies from ten iterations of three migration events. The variance of relatedness explained by this model was estimated to be 99.98%. The line represents gene flow; the direction of the arrow is the direction of gene flow; the color bar shows the migration weight, with red representing strong gene flow and yellow representing weak gene flow. The direction of gene flow was PE into ML, LC into CY1 and ML into CY2. (**e**) Residuals matrix for migration model.

### 3.4. Gene Flow Analysis

In order to infer the split pattern and mixtures among populations of *T. doichangensis*, we used allele frequencies from six populations to infer admixture events. Supplementary Figure S4 shows the maximum likelihood value and variance explained for each event based on *OptM* package analysis in TreeMix, which was used to choose migration events varying from 0 to 6. The model without migration events explained 99.71% of the variance, with the highest score for the variance (99.98%) being achieved when m = 3; adding more migration events did not explain more variance. Figure 5d shows the optimal "migrate" result for the last iterations. The result indicated that the direction of gene flow was PE into ML, LC into CY1 and ML into CY2. Although TreeMix detected no strong signal of migration between CY1 and CY2, the residues showed moderate gene flow between CY1 and CY2 (Figure 5e). The ML into CY2 inferred migration event did not match geographical patterns, indicating a special flow channel between the populations, which affected the introgression. We then used the three-population and four-population test (f3-statistic/f4-statistic) to evaluate the statistical significance of the mixtures (Tables S1 and S2). The three-population test indicated that a mixture existed between populations, in which CY1 was the mixing result of multiple populations (Table S1). The four-population test confirmed that there was greater gene flow between the two populations CY1 and CY2 in Cangyuan county with a relatively high mixing degree (Table S2), which also suggested low genetic differentiation between CY1 and CY2. Meanwhile, the TreeMix results were corroborated by the four-

population tests. Furthermore, the f4-statistic results showed that there was lower gene flow between most populations, and it has been suggested that the existence of gene flow can maintain the stability of small populations [60]. Although we did not add outgroups to the analysis, the results can still be used as a reference for relations of populations.

## 4. Discussion

### 4.1. Genetic Diversity

The genetic variation of species is the premise of local adaptation and evolution, and it is also considered an important parameter to determine the priority of population conservation in the protection of endangered plants [12,61]. According to the CBD, genetic diversity is one of the three basic elements of biodiversity, and it is the focus of many conservation genetics studies [62]. Therefore, genetic diversity is one of the most important values in assessing biodiversity for conservation.

When studying the genetics of wild populations, it is desirable to sample tens, hundreds or even thousands of individuals. However, there are some rare and endangered taxa with narrow distributions. The genetic diversity of narrowly distributed species is likely to be lower than widespread plant species because of inbreeding depression and genetic drift [16,63]. A previous study on the genetic diversity of *T. doichangensis* based on random amplified polymorphic DNA indicated high genetic differentiation, a low level of genetic diversity and a poor gene flow [35]. Ramanatha and Hodgkin [64] stated that different molecular markers show different levels of genetic diversity and, unexpectedly, GBS indicated that the nucleotide genetic diversity of *T. doichangensis* is high (Figure 3; Table 2). In all populations, the results for the diversity parameters for *T. doichangensis* populations utilizing the expected heterozygosity (*He*), observed heterozygosity (*Ho*) and nucleotide polymorphism (*Pi*) indicated that ML and XSBN had higher genetic diversity (Figure 3). Moreover, the number of private alleles was the highest in XSBN (Table 2). The genetic diversity of species is generally influenced by their distribution range, life history, breeding system, seed dispersal mechanism and evolutionary history [16]. According to previous studies, *T. doichangensis* has a low germination rate and low fruiting rate in nature, thus limiting the development of the population [65]. Furthermore, *T. doichangensis* is distributed in evergreen broad-leaved forests where sunlight cannot reach the ground through the tree layer, and the dense trees increase the mortality of young *T. doichangensis* seedlings competing for sunlight and space to survive [66]. This is one of the reasons for the lower genetic diversity (*He* = 0.2232–0.3059; Table 2) compared to other genera in the family; e.g., *Castanopsis* (EST-SSR, *He* = 0.644; [67]) or *Quercus castanea* (nSSR, *He* = 0.762; [68]). *T. doichangensis* is also facing the dual impact of biological invasion and human activities, and its habitat has also suffered serious damage. Shrinking habitats, lower population sizes, and fewer individuals in species seriously affect genetic diversity [35]. In this study, the populations of *T. doichangensis* still retained a certain degree of genetic diversity (Table 2; Figure 3), which showed that populations of *T. doichangensis* still have a certain ability to resist external risks. However, in species protection, only relying on the species itself to resist the adverse environment is not desirable. Accelerated habitat fragmentation and population size reduction can exacerbate genetic drift and inbreeding, resulting in rapid loss of genetic diversity.

In addition, genetic diversity is usually higher in outcrossing species than in selfing species [69,70]. Notably, inbreeding coefficients ($F_{IS}$) of all populations were negative (Table 2; Figure 3a), implying that there is no inbreeding in the populations of *T. doichangensis*, thus increasing the genetic diversity. We assume that, in the past few decades, with the rapid reduction of the population size, the genetic drift effects have not yet accumulated, and the species are not strongly affected by genetic drift or inbreeding and retain relatively high heterozygosity and genetic variation in limited populations.

### 4.2. Genetic Differentiation, Genetic Structure and Gene Flow

Genetic differentiation is often strongly influenced by selection pressure, gene flow and life history [71]. For endangered populations, isolated populations and broken habitats are highly susceptible to producing high genetic differentiation [72]. The $F_{ST}$ values of *T. doichangensis* ranged from 0.0462 to 0.1402, with an average $F_{ST}$ value of 0.0921 (Table 3), revealing that moderate genetic differentiation among the populations occurred according to Wright's theory (Wright, 1965). However, it is generally accepted that gene flow can block genetic differentiation due to genetic drift when $Nm > 1$ [73]. We calculated that $Nm = (1 - F_{ST})/4 * F_{ST} = 2.7194$ in the *T. doichangensis* populations, which implied that there was gene flow among the populations counteracting genetic differentiation [73,74]. Furthermore, AMOVA showed that 88.24% of the genetic variation occurred within *T. doichangensis* populations but not among populations (Table 4), while the $F_{IS}$ results showed that the inbreeding coefficients of all the populations of *T. doichangensis* were negative (Table 2), indicating that there was no inbreeding within populations and, thus, genetic differentiation was lower [23]. For rare and endangered plants, population genetic differentiation is usually affected by large geographical distances between sampled populations [69,70]. However, the genetic distance and geographic distance between populations of *T. doichangensis*, the Admixture analyses, the PCA and the gene flow demonstrated that the populations of *T. doichangensis* are not genetically separated in accordance with their geographical distributions (Figures 4b and 5). The previous finding that pairwise genetic distances between populations were not correlated with geographical distances was supported by the observation that one of the Chinese populations is most similar to the Thai population [75].

The distance among the populations in this study was about 50 km, which hardly allows gene flow (pollen or seeds) between populations. However, there were three gene flow events among the six populations in TreeMix; i.e., PE to ML, LC to CY1 and ML to CY2 (Figure 5d). Three-population and four-population tests also indicated partial gene flow between populations (Tables S1 and S2). Admixture, PCA and DAPC showed that the optimal cluster of populations was 3, and the genetic structure of some populations was mixed (Figure 5). All the above results indicated that there was some gene flow among the populations of *T. doichangensis* and that their genetic structure is not strongly geographically heterogeneous. At the same time, we found that Admixture, PCA and DAPC all pointed to the possibility that the more independent genetic components for XSBN and ML populations may retain more ancestral information, while the PE populations may have gradually separated due to river barriers after dispersal.

Previous studies found that most of the seed and seedling traits of *T. doichangensis* showed significant differences among populations [75], confirming the hypothesis that *T. doichangensis* possesses very strong genetic differentiation within its populations. In addition, in line with our observations, seedlings grow sporadically around the parent tree. Meanwhile, Sun [35] noted the presence of small beetles in the male flowers of *T. doichangensis*, so it can be presumed that pollinator beetles may be active. However, the flowers of *T. doichangensis* are small and dull in color, so they cannot attract more pollinating insects, and pollination by beetles will cause *T. doichangensis* pollen to spread over a short distance and make the gene flow between populations difficult, leading to differences between populations. Hence, the differentiation among the populations could be attributed to distance-limited pollen flow and short-distance seed dispersal. We suggest that habitat destruction and excessive exploitation may have led to low gene flow, which in turn resulted in the differences in seed and seedling morphological traits among populations.

### 4.3. Conservation Significance

*Trigonobalanus doichangensis* is a rare and endangered fagaceous plant with evolutionary significance for the understanding of the phylogeny and biogeography of Fagaceae and even Chinese flora more generally. It is currently restricted to a few sites in Yunnan province in southwestern China, as well as one in northern Thailand [35,75]. *T. doichangensis*

was placed on the national Rare and Endangered Species List of China in 1984 because of its limited distribution and the destruction of its habitat within China [66,76]. *T. doichangensis* is China's second-ranked taxon for priority of national protection [36] because of its endangered status and its scientific value. However, the rare and endangered *T. doichangensis* is exposed to the risk of extinction due to the destruction and degradation of forest habitats, agriculture, silviculture and the harvesting of wood for fuel and tool making [75].

The disappearance of rare plants prevents the tracking of the evolutionary and biogeographic history of plants, which not only has consequences for medicinal and other economic uses but also limits the potential resources about past climate change and future implications [77]. In addition, although only a small number of plant species have been exploited by humans, many others play important roles in natural ecosystems, and rare species may also have novel traits that could be useful in the future. Thus, the impacts of economic development on rare plants and their habitats need to be recognized and addressed before these potential resources are lost forever.

The genetic diversity of *T. doichangensis* obtained through GBS has important conservation significance for narrowly distributed species. The most effective approach to conserving endangered species is in situ conservation [25]. A previous study based on RAPD suggested that conservation of this species should include preservation of each extant population [75]. Seedlings and saplings are rarely found in wild populations of *T. doichangensis*, and its habitat has been disturbed and destroyed as a result of various factors, including insects and low seed-germination percentages [34,35]. Therefore, it is imperative to establish conservation plots to protect its natural habitat. In accordance with its genetic diversity and structure, we suggested that ML and XSBN should be protected as in situ conservation priorities as soon as possible so they can be used as germplasms with high diversity. Moreover, because moderate genetic differentiation and certain genetic structure were revealed among populations—particularly, the gene flow in the ML and XSBN populations—germplasm resources from each population should be not mixed and instead should be used separately to avoid the risk of outbreeding depression. Considering that the habitats of *T. doichangensis* have been devastated and fragmented by human activity, and that the seed setting percentage is low and seedling growth is limited, artificial seedlings should also be raised and transplanted to areas with similar habitats in order to expand the population size and protect the gene pool of the species. In addition, conservation campaigns should be conducted in towns and villages around the distribution area of *T. doichangensis* to protect the original habitat and restore the function of the surrounding plant community.

### 5. Conclusions

In summary, we used the high density of SNP loci generated by GBS for population genomic analysis of *T. doichangensis* with 39 individuals from 6 populations. Our data indicated that there was high genetic diversity and moderate genetic differentiation in the six populations of *T. doichangensis*. At the same time, most of the genetic variation in *T. doichangensis* occurred among populations, and there was some gene flow among populations to counteract the genetic differentiation caused by genetic drift. Based on these results, we propose a strategy for in situ conservation of the ML and XSBN populations of *T. doichangensis* with high genetic diversity and highlight the importance of germplasm collection, artificial seedlings and conservation promotion, providing an important reference and guidance for the conservation of *T. doichangensis* populations and similarly endangered species.

**Supplementary Materials:** The following supporting information can be downloaded at: https://www.mdpi.com/article/10.3390/d14080666/s1, Figure S1: The results of cross-validation error (a) and ancestry composition when K = 2 to K = 6 in Admixture analysis (b); Figure S2: The results of PCA based principal component 1 to principal component 3 (PC1 and PC2 and PC3); Figure S3: Variance explained by PCA eigenvalues (a) and optimal K-value results based on BIC (b) in DAPC

analysis; Figure S4: *OptM* results for TreeMix when run from 0 - 6 migration event(s); Table S1: Three-population test for *T. doichangensis*; Table S2: Four-population test for *T. doichangensis*.

**Author Contributions:** Conceptualization, L.H., H.-H.M., Y.C. and J.L.; Funding acquisition, H.-H.M. and Y.C.; Project administration, H.-H.M. and J.L.; Visualization, L.H. and H.-H.M.; Formal analysis, L.H.; Writing—original draft, L.H. and H.-H.M.; Writing—review and editing, L.H., X.-G.L., S.-S.Z., C.-Y.Z., Y.-H.T., Q.R., H.-H.M., Y.C. and J.L. All authors have read and agreed to the published version of the manuscript.

**Funding:** This research was supported by funding from the Southeast Asia Biodiversity Research Institute, Chinese Academy of Sciences (No. Y4ZK111B01), to H.-H.M.; from the CAS "Pioneer Hundred Talents" Program awarded to Y.C.; from the Youth Innovation Promotion Association, Chinese Academy of Sciences (No. 2018432), to H.-H.M.; and from the CAS "Light of West China" Program to H.-H.M.

**Institutional Review Board Statement:** Not applicable.

**Data Availability Statement:** The Data and Code supporting the current study is available from the first author (Ling Hu; huling20@outlook.com or huling@xtbg.ac.cn) on requeste.

**Acknowledgments:** We thank Jian-Hua Xiao and Chao-Nan Cai for help in the GBS and AMOVA. We also thank Xiao-Yan Zhang for kind help in using ArcGIS. Thanks to Chu-Meng Zhu, Huan-Xing Ren and Liu-Qing Yang for help in using R and Linux.

**Conflicts of Interest:** The authors declare no conflict of interest.

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
