# Peer review of "Conservation Significance of the Rare and Endangered Tree Species, Trigonobalanus doichangensis (Fagaceae)"

_diversity, doi:10.3390/d14080666_

Round 1
Reviewer 1 Report
In general, the manuscript to be well written and of interest. The present study investigated the genetic diversity of T. doichangensis to provide information on the effective preservation of the genetic diversity of the species based on a high density of SNPs loci generated by the GBS-seq technology. This paper indicates that there is high genetic diversity and moderate genetic differentiation in the six populations of T. doichangensis and it provides us with detailed preservation strategies of these populations.
I have a few comments/questions for the authors:
1. Maybe it is better to add the phylogenetic analysis to understand the evolutionary relationships of the populations.
2. How about the population demographic history of T. doichangensis? The sentence (in P12 L437) “At the same time, we found that Admixture, PCA, and DAPC all point to more independent genetic components for XSBN and ML populations may retain more ancestral information……” likely mentioned some but not clear.
3. P5: The header font is not consistent in Table 1.
4. Please use the consistent format of journal names in references, for example:
P14 L525 and L545: PLoS one and PLoS ONE is not consistent.
P14 L550: J Appl Ecol. should be J. Appl. Ecol.
P16 L631 and L642: Tree Genet Genomes and Tree Genetics & Genomes should be Tree Genet.
Author Response
Response to Reviewer#1
Reviewer#1: In general, the manuscript to be well written and of interest. The present study investigated the genetic diversity of T. doichangensis to provide information on the effective preservation of the genetic diversity of the species based on a high density of SNPs loci generated by the GBS-seq technology. This paper indicates that there is high genetic diversity and moderate genetic differentiation in the six populations of T. doichangensis and it provides us with detailed preservation strategies of these populations.
I have a few comments/questions for the authors:
Response: We thanks Reviewer for the detailed comments, and they are importantly for improving our manuscript. Below, we give our responses to the comments and question.
Maybe it is better to add the phylogenetic analysis to understand the evolutionary relationships of the populations.
Response: Thanks for your comments, it is very important for the phylogenetic analysis to understand the evolutionary relationships of the populations. However, in this study, we focus more on the genetic diversity and conservation aspects, so the phylogenetic aspects will be carried out in the following study.
How about the population demographic history of T. doichangensis? The sentence (in P12 L437) “At the same time, we found that Admixture, PCA, and DAPC all point to more independent genetic components for XSBN and ML populations may retain more ancestral information……” likely mentioned some but not clear.
Response: Thanks for the comments. Indeed, as we mentioned in this study, we found that Admixture, PCA, and DAPC all point to more independent genetic components for XSBN and ML populations may retain more ancestral information, while the PE populations could have gradually separated due to river barriers after dispersal. However, this study focused on conservation strategies through the assessment of genetic diversity and the exploration of the genetic structure of T. doichangensis, we have not studied more population demographic history of T. doichangensis. And, we have revised the related section in the revised manuscript.
P5: The header font is not consistent in Table 1.
Response: Changes have been changed in the revised manuscript. Thanks for your comment.
Please use the consistent format of journal names in references, for example:
P14 L525 and L545: PLoS one and PLoS ONE is not consistent.
P14 L550: J Appl Ecol. should be J. Appl. Ecol.
P16 L631 and L642: Tree Genet Genomes and Tree Genetics & Genomes should be Tree Genet.
Response: We have checked all format of journal in the references and above questions have been done in the revised manuscript. Thank you for your comment.
Reviewer 2 Report
Dear authors
Is it known how gene flow occurred after glaciation? Antoine Kremer et al (Kremer, A., Ronce, O., Robledo-Arnuncio, J. J., Guillaume, F., Bohrer, G., Nathan, R., ... & Schueler, S. (2012). Long-distance gene flow and adaptation of forest trees to rapid climate change. Ecology letters, 15(4), 378-392.) and (Gerber, S., Chadœuf, J., Gugerli, F., Lascoux, M., Buiteveld, J., Cottrell, J., ... & Kremer, A. (2014). High rates of gene flow by pollen and seed in oak populations across Europe. PloS one, 9(1), e85130.) demonstrated such gene flow for oak, for example, for which 3 refugia were described (in the Iberian Peninsula, the Apennine Peninsula and the Balkans). In the case of T. doichangensis populations, refugia could represent a source of biodiversity of particular importance for the conservation of the species.
Has poor seed production been observed or is it due to inbreeding? Also, did the seeds collected and sown germinate at a low rate?
Were the seeds and seedlings from the least genetically diverse populations of low vigour and susceptible to fungal diseases? How do the authors assess the health status of the populations studied? Does low genetic diversity affect susceptibility to diseases caused by insect pests and fungi?
Carefully reread the text for typos, e.g., L92 is missing the "s" next to "species." Unify the spelling of characters, e.g. %, sometimes they are placed after the space, e.g. L248 (88.52%), sometimes the opposite, e.g. L250.
In L249 GC% should better be replaced by GC "percentage", especially since the number 88.52% is given directly.
Author Response
Response to Reviewer #2
Is it known how gene flow occurred after glaciation? Antoine Kremer et al (Kremer, A., Ronce, O., Robledo-Arnuncio, J. J., Guillaume, F., Bohrer, G., Nathan, R., ... & Schueler, S. (2012). Long-distance gene flow and adaptation of forest trees to rapid climate change. Ecology letters, 15(4), 378-392.) and (Gerber, S., Chadœuf, J., Gugerli, F., Lascoux, M., Buiteveld, J., Cottrell, J., ... & Kremer, A. (2014). High rates of gene flow by pollen and seed in oak populations across Europe. PloS one, 9(1), e85130.) demonstrated such gene flow for oak, for example, for which 3 refugia were described (in the Iberian Peninsula, the Apennine Peninsula and the Balkans). In the case of T. doichangensis populations, refugia could represent a source of biodiversity of particular importance for the conservation of the species.
Response: Thank you for the constructive and detailed comments, and they are important to improve the manuscript. Below, we give our responses to your comments and questions.
Global cooling was widespread during the Miocene to Pleistocene glacial period, a change that had a dramatic impact on species differentiation at high latitudes while causing tropical-subtropical components to retreat to lower elevations, thereby promoting species habitat connectivity and increasing gene flow (Chattopadhyay et al., 2017). As climate changes, post-ice age temperature rebound and sea level rise accelerates species distribution dispersal on land and islands, while large fluctuations in climate and habitat fragmentation will result in impaired gene flow over long distances (Kremer et al., 2012). Plant refugia are the starting point for post-ice age redistribution, and repeated climate fluctuations and sea level changes will result in dramatic changes in the distribution patterns and genetic structure of species, with many populations experiencing reduced genetic diversity due to founder effect and bottleneck effect, while populations in refugia areas will retain more genetic diversity (Chen et al., 2011).
We agree with you that T. doichangensis populations may be affected by refugia and thus retain some genetic diversity. However, we focus more on the assessment of genetic diversity and conservation measures, the corresponding work on changes in populations of T. doichangensis during and after the glacial cycle and the response of populations to climate change will be our main task in the subsequent work.
Has poor seed production been observed or is it due to inbreeding? Also, did the seeds collected and sown germinate at a low rate?
Response: Previous study about T. doichangensis seeds from the Menglian, Lancang, Ximeng, and Canyuan areas within the natural range of T. doichangensis revealed a low seed production rate with only 9.8% (Zhou et al., 2003). However, our study showed negative inbreeding coefficients for all populations (Table 2; Figure 3b), suggested that inbreeding between populations of T. doichangensis is almost non-existent, and that the species is not strongly affected by inbreeding. In addition, it was found that seed germination rates for T. doichangensis were generally low, around 6.33~7.50%, and received high effects from temperature and rainfall (Zhou et al., 2003). In general, seeds germinate easily to form seedlings under suitable conditions of temperature and soil moisture content, and field surveys have found more seedlings growing in sunny areas of the forest, while in natural habitats with a large community cover, it is difficult to find seedlings growing even though there are a large number of mature fruits. In addition, T. doichangensis usually flowers from October to November and the fruit ripens around March of the following year, when the season of low temperatures and low precipitation in its native habitat, temperature and humidity are very unfavorable to the germination of seeds.
Were the seeds and seedlings from the least genetically diverse populations of low vigour and susceptible to fungal diseases?
Response: Thank you very much for providing a thought-provoking idea, low vigour and susceptible to fungal diseases are important for seed and seedling selection. Your comments also provide important suggestions for our future research work.
How do the authors assess the health status of the populations studied?
Response: T. doichangensis is distributed in southwest and southern Yunnan, China. Based on these evidences from our work and other works, we believe that in the range of Yunnan, China, there is a concentration of indigenous peoples of Dai, Wa and Laku, who are largely dependent on these mountain resources for their livelihoods and therefore present serious persecution and threat to the habitat of the T. doichangensis. T. doichangensis is used for agricultural tools and house construction, as well as for fuel wood, and these unreasonable uses have led to the loss of germplasm and limited population development. In addition, with the development of socio-economic and population numbers, more and more cash crops (such as sugar cane, tea, and lychee) have come into the local population's attention, which has led to a serious impact on the habitat of the T. doichangensis and the fragmentation of its natural habitat, seriously threatening the survival and development of its populations. On the other hand, the natural range of the T. doichangensis is also under serious threat from invasive alien species, Eupatorium odoratum and Ageratina adenophora were originally introduced from Myanmar from the southwest of Yunnan, and their invasion route is the natural range of the T. doichangensis. Due to the continuous expansion and exclusion of invasive species, the population structure and diversity of T. doichangensis has also changed significantly, and the rapid reproduction of invasives in T. doichangensis communities has resulted in the inability of T. doichangensis seeds to grow normally in the communities, seriously affecting the natural regeneration of T. doichangensis.
In addition, this work found that populations of T. doichangensis still retain a degree of genetic diversity (Table 2; Figure 3a), suggesting that T. doichangensis populations currently still have some resilience to external risks. Therefore, the current population of T. doichangensis is at an important critical period, where excessive anthropogenic disturbance and habitat fragmentation have reduced its population size and restricted its growth, and without effective conservation, the genetic diversity of T. doichangensis will rapidly decrease, thus accelerating the loss of germplasm resources.
Does low genetic diversity affect susceptibility to diseases caused by insect pests and fungi?
Response: Genetic diversity is one of the key pillars of biodiversity, and high genetic diversity increases the ability to survive in the wild plant and reduces the risk of extinction and will enable effective use in the improvement of cultivated species, while the different characteristics of different cultivars of the same species, such as growth patterns, resistance to pathogens, tolerance to adversity and productivity, are all important for the genetic gene pool (Esquinas-Alcázar, 1993; Phillips et al., 2012; Zhang et al., 2019). In addition, genetic diversity is also an important factor in plant resistance to climate change and defense against insect pests and diseases (Esquinas-Alcázar, 1993). The more homogeneous the genetic diversity, the more vulnerable it will be to drought, viruses, insect pests and fungi. The greater the genetic diversity, the greater will be the evolutionary potential and resistance to environmental and disease pests (Esquinas-Alcázar, 1993).
Carefully reread the text for typos, e.g., L92 is missing the "s" next to "species." Unify the spelling of characters, e.g. %, sometimes they are placed after the space, e.g. L248 (88.52%), sometimes the opposite, e.g. L250.
Response: Thanks for your comment, we have checked the manuscript for typos and misspellings and have corrected them in the revised manuscript.
n L249 GC% should better be replaced by GC "percentage", especially since the number 88.52% is given directly.
Response: Changes have been done in the revised manuscript. Thanks for your comment.
Mainly References:
Chattopadhyay, B.; Garg, K.M.; Gwee, C.Y.; Edwards, S.V.; Rheindt, F.E. Gene flow during glacial habitat shifts facilitates character displacement in a Neotropical flycatcher radiation. BMC Evol. Biol. 2017, 17, 1-15
Chen, D.M.; Kang, H.Z.; Liu, C.J. An overview on the potential quaternary glacial refugia of plants in China mainland. Bull. Bot. Res. 2011, 31, 623-632.
Esquinas-Alcázar, J.T. Plant genetic resources. Chapman & Hall, London, UK, 1993; pp. 33–51.
Kremer, A.; Ronce, O.; Robledo‐Arnuncio, J.J.; Guillaume, F.; Bohrer, G.; Nathan, R.; Bridle, J.R.; Gomulkiewicz, R.; Klein, E.K.; Ritland, K.; et al. Long‐distance gene flow and adaptation of forest trees to rapid climate change. Ecol. Lett. 2012, 15, 378-392.